# Review of the Electrospinning Process and the Electro-Conversion of 5-Hydroxymethylfurfural (HMF) into Added-Value Chemicals

**DOI:** 10.3390/ma15124336

**Published:** 2022-06-19

**Authors:** Maximilien Coronas, Yaovi Holade, David Cornu

**Affiliations:** Institut Européen des Membranes, IEM UMR 5635, Univ Montpellier, ENSCM, CNRS, 34090 Montpellier, France; maximilien.coronas@enscm.fr

**Keywords:** electrospinning, gold, polyacrylonitrile, electrosynthesis, thermal treatment, hydrogen, 5-hydroxymethylfurfural

## Abstract

Electrochemical converters (electrolyzers, fuel cells, and batteries) have gained prominence during the last decade for the unavoidable energy transition and the sustainable synthesis of platform chemicals. One of the key elements of these systems is the electrode material on which the electrochemical reactions occur, and therefore its design will impact their performance. This review focuses on the electrospinning method by examining a number of features of experimental conditions. Electrospinning is a fiber-spinning technology used to produce three-dimensional and ultrafine fibers with tunable diameters and lengths. The thermal treatment and the different analyses are discussed to understand the changes in the polymer to create usable electrode materials. Electrospun fibers have unique properties such as high surface area, high porosity, tunable surface properties, and low cost, among others. Furthermore, a little introduction to the 5-hydroxymethylfurfural (HMF) electrooxidation coupled to H_2_ production was included to show the benefit of upgrading biomass derivates in electrolyzers. Indeed, environmental and geopolitical constraints lead to shifts towards organic/inorganic electrosynthesis, which allows for one to dispense with polluting, toxic and expensive reagents. The electrooxidation of HMF instead of water (OER, oxygen evolution reaction) in an electrolyzer can be elegantly controlled to electro-synthesize added-value organic chemicals while lowering the required energy for CO_2_-free H_2_ production.

## 1. Introduction

Twenty years ago, Professor Peter William Atkins, one of the best chemistry educators of our era, considered that we need to broaden our view of what electrochemistry is; it is not just about electrode potentials and electrolysis, this is half of modern chemistry; electrochemistry―in the broadest sense―will be one of the great achievements in chemistry in the next millennium, and we need prepare people for it [1]. Indeed, for centuries, fossil fuel resources have played an essential role in the industry, including chemical and petrochemical manufacturing, making them central to us nowadays. Due to the current environmental situation, where CO_2_ levels have risen rapidly, there is an increasing demand for energy, and there is a decrease in fossil fuel feedstock, a lot of research has been dedicated to the development of alternative processes [2,3,4].

Current developments are oriented towards producing less waste by making organic chemicals in a “greener” fashion from biomass derivates (for example, avoiding the use of organic solvents, working under room temperature, having better control of coproducts, etc.) [2,3,4,5,6]. For instance, several candidates such as 2,5-diformylfuran (DFF), 5-hydroxymethyl-2-furancarboxylic acid (HFCA), 5-formyl-2-furancarboxylic acid (FFCA), and 2,5-furandicarboxylic acid (FDCA) have been generated by the oxidation of 5-hydroxymethylfurfural (HMF) under different conditions [2,7,8,9]. HMF derives from cellulose and hemicellulose, the two primary constituents of lignocellulosic biomass (60–75%), and has the potential to be converted into platform biofuels [10,11]. Specifically, the biosourced compound HMF is an intermediate for the synthesis of bio-renewable FDCA, which is the monomer for the polyethylene furanoate (PEF) biopolymer materials of industrial relevance as it is a green alternative to polyethylene terephthalate (PET) [10,12,13]. On the other hand, by producing these value-added chemicals from an electro-oxidation process in an electrolyzer, the electro-reduction step could be used to create another value-added chemical. For example, the electrocatalytic reduction of water (in alkaline media) or protons (in acidic media) is considered to be an ideal process for generating H_2_ without by-products and does not lead to the emission of polluting species, unlike the current methods, which produce a large amount of CO_2_ [3,14,15]. By using this coproduction method, the use of fossil fuels for energy production can be gradually replaced by generating H_2_ from renewable resources (H_2_O) and for the chemical industry by using the abundant biomass. However, for this electrochemical biomass-to-hydrogen technology, further efforts are needed to overcome the kinetics and/or the selectivity of both anode and cathode half-cell reactions. This will reduce the price of H_2_ production and make the technology competitive with current methods based on the thermal decomposition of fossil fuels [16,17,18].

To be used in electrochemical reactors, the catalytic materials (ideally in the form of nanoparticles in order to reduce the total amount and to regulate the kinetics) must be immobilized on an electrically conducting support. Therefore, the need for an increased reaction yield and selectivity by minimizing the waste production resulted in the development of novel catalytic supports. Atomic layer deposition (ALD) has presented the possibility to design supports for catalytic reactions; however, this method can be expensive and time consuming, and it is quite impossible to dope the interior of the fibers by metallic species [19,20,21,22,23]. Moreover, electrospinning is a versatile technique for generating ultrathin fibers that can lead to catalytic supports. Metal particles can be added on the surface and inside the fibers. Different types of spinning methods have been explored in the past decades, for example, wet spinning, dry spinning, melt spinning, and gel spinning. However, the fibers could not reach the sub-micrometer scale. Then, electrospray and electrospinning were discovered by adding voltage in order to create ultra-thin continuous fibers, leading to the possibility of reaching the sub-micrometer scale [5,24,25,26,27,28,29].

Different polymers have been tested in electrospinning, such as polyvinylpyrrolidone (PVP) [30,31,32], polylactic acid (PLA) [33], polyacrylonitrile (PAN) [34,35], poly(ethylene oxide) (PEO) [31], etc. [5,24,25,26,36], and the results are promising. However, further investigations are needed now to fabricate advanced conducting materials that could lead to electrode materials for the electrosynthesis of value-added chemicals. A recent advancement in the method has shown that it is possible to add metallic species into the polymeric fiber matrix and then thermally convert the electrospun mat into nanometer sized catalysts with metallic species inside and outside each microfiber. To perform such a synthesis for an electrically conducting material (to act as a support in electrochemical applications) and an electrocatalytically active material (to act as active site in heterogeneous electrocatalysis), a suitable procedure of thermal treatment is needed. To achieve that from a raw PAN-based polymer modified by metallic cations, two steps need to be followed: stabilization and the carbonization [37]. Their tight control enables one to fabricate a free-standing catalytic support for the implementation in electrochemical fields. The details are explained later in this review. Currently, the main precursor of the electrospun carbon fiber (CF, also referred to as carbon nanofiber (CNF) even if the diameter of the fiber is mostly higher than 100 nm) is the PAN; 90% of carbon fibers is made from PAN [5,35,38].

The aim of this review is to summarize the latest developments regarding the use of electrospinning to design advanced electrocatalytic materials. We will present the different steps starting from PAN solutions, the parameters of the electrospinning process, the impact of metallic species, and the thermal treatment. Then, the electrocatalysis of the HMF will be discussed along with the electrocatalytic characterizations.

## 2. Synthesis of PAN Fibers and Gold Nanoparticles

### 2.1. Formulation of a Suitable Electrospinning Solution

PAN is the most used polymer for the preparation of CNFs [5,35,38]. It offers the possibility to add different metals [32,39,40] or molecules [27,41,42,43] to the fibers during the synthesis process. Like the other polymers (PVP, PLA, PEO, etc.), PAN is commercially available. The commonly used solvents to dissolve PAN are dimethyl sulfoxide (DMSO) and dimethylformamide (DMF). Although DMF has side effects (it is carcinogenic, flammable, and harmful), it is widely used to solubilize the polymer. DMSO and DMF could both be used for the electrospinning process because they provide the two desired criteria: the ability to solubilize the polymer and evaporate during the electrospinning process [34]. As summarized by Pastoriza-Santos and Liz-Marzan, under appropriate conditions, DMF can also act as a reducing agent and lead to metal nanoparticles, mainly Au and Ag [30]. Different studies have reported the preparation of PAN solutions in DMF with or without adding metals for the electrospinning such as Zhang et al. [35], Holade et al. [39], and Both Engel et al. [34]. The formulation of a suitable electrospinning solution is very critical for the electrospinning itself. It is important to ensure that the solution is as homogeneous as possible and that it does not change significantly during the electrospinning process, which can take several hours. The majority of the studies used PAN with a molecular weight of 150,000 g/mol [44]. This makes it easier to reproduce the experiments and to have a similar background of information, which is not the case for the other parameters that will be reviewed below. The key parameter is the percentage of PAN in the electrospinning solution to avoid having a mixture that is neither too liquid nor too viscous and therefore difficult for electrospinning. Moreover, the addition of an inorganic or organic compound can impact the stability of the mixture, as well as the electrospinning conditions. The reported amount ranges from 3 wt.% [45] to 14 wt.% [46], while the majority of the studies are done for 10–11 wt.% of PAN [34,39,40,47,48,49,50,51,52]. The underlying parameter is the true volume of PAN solution loaded in the syringe (see Figure 1a). The reported values are 7.2 mL [39], 4 mL [34], and 2 mL [53]. We note that not all of the prepared volume of the solution is always used; it depends on the type of desired material. This key parameter is not specified in the majority of studies, which compromises the repeatability and replicability of the results because the total amount of electrospun mat logically depends on the used volume for a fixed concentration.

Even if the formulation duration, or the temperature, are not always fully disclosed in all studies, the duration is 12 h at 60 °C [56], 3–6 h at 70 °C [34,39], and 2 h at 80 °C [47] (50 °C without an indication of the used duration [46]). These differences between the used conditions to prepare the electrospinning solution may come from the intended end-use, the addition of other chemical species (for example, metallic species), or the type of solvent for the dissolution of PAN. After ending the formulation and preparation of the solution, the mixture is loaded in a syringe (see Figure 1a) and the electrospinning process can properly start.

### 2.2. Electrospinning Process

Electrospinning is routinely conducted at room temperature [57] and is recognized as an efficient technique for the fabrication of 1D nanofibers [17,58]. The diameter of the constituting fibers ranges from micrometers (e.g., 10–100 µm) to sub-microns (and even nanometers) [32,58]. It is a cost-effective technology that generates non-woven fibers with a high ratio of surface-area-to-volume and tunable porosity. As shown in Figure 1b, the electrospinning involves an electro-hydrodynamic process, whereby a liquid droplet is electrified to generate a jet, followed by stretching and elongation to generate fibers [55]. An electrostatic force should be created in order to produce polymer filaments, as shown in the Figure 1 [58,59]. Furthermore, a DC voltage in the range of several tens of kVs is necessary to run the electrospinning. There are basically three components needed to fulfill the process: a high-voltage supplier, a capillary tube with a pipette or needle of small diameter, and a metal collecting screen, as shown in Figure 1 [5,57,58]. Over the years, more than 200 polymers have been electrospun for various applications, and more than 50 different polymers have been successfully electrospun into ultra-fine fibers, as represented in Table 1 [55,57,58].

The electrospinning process is relatively simple. Before reaching the electrically conducting collector, the jet solution evaporates or solidifies and is collected as an interconnected network of fibers. Mutual charge repulsion and the contraction of the surface charges of the counter electrode cause a force directly opposite to the surface tension. When the applied electrostatic forces overcome the fluid surface tension, the electrified fluid forms a jet out of the capillary tip towards a grounded collecting screen. At a high rotation speed up to thousands of rpm (revolutions per minute), electrospun nanofibers can be oriented circumferentially [5,32,58,59,60]. To summarize, the process consists of three stages: (i) jet initiation and the extension of the jet along a straight line; (ii) the growth of whipping instability and the further elongation of the jet, which may or may not be accompanied with the jet branching and/or splitting; and (iii) the solidification of the jet into fibers.

During an electrospinning process, there are few parameters that need to be controlled to ensure that the experiments meet the three criteria of being highly repeatable, reproducible, and replicable. Electrospun fibers and the control of their diameters are largely determined by the processing parameters, including [5,32,57,58,59,60]: (i) solution parameters such as viscosity, elasticity, conductivity, surface tension, and polymer-solvent affinity; (ii) process parameters such as hydrostatic pressure in the capillary tube, electric potential at the capillary tip, the gap (the distance between the tip and the collector), and the feed rate; and (iii) ambient parameters such as solution temperature, humidity, and air velocity in the electrospinning chamber.

Furthermore, a dopant such as metal or alloy could be added in the electrospun solution. If the dopant increases the conductivity of the polymer solution, it will make the nanofiber thinner [32]. Each of these parameters can affect the electrospinning process—by harming the fibers’ morphology, for example. Hence, by a proper control of these parameters, one can fabricate electrospun fibers with desired morphologies and diameters. Further information about the impact of the different parameters can be found in ref. [60].

### 2.3. Formation and Synthesis of Nanoparticles

There are different ways to synthesize NPs, for example, the constant potential electrolysis [61], the annealing of a salt precursor coated on a surface [5], the use of a reducing agent [39,62], ultra-violet (UV) light [46,47], the phase transfer [16,63], the sol-gel method [40], or directly by electrospinning. Additionally, different methods can be combined as reported by Anka et al. [47] and Sawada et al. [46]. In their studies, they formed AuNPs by electrospinning under UV light, reactions of photo-polymerization, and photo-crosslinking. UV, microwave, thermal treatment requires extra energy to produce the AuNPs. A study made by Both Engel et al. [34] showed that it is possible to create metallic particles and specifically nanostructured gold particles on carbon fibers by the electrospinning process and prevent impurities that are usually present on the surface of AuNPs electrodes prepared by chemical vapor deposition (CVD). By the method of Both Engel et al. [34], small AuNPs are formed inside the fibers and play a role in the physical properties of the carbon electrodes, whereas the large ones are on the surfaces of the fibers to act as active sites during the electrocatalytic reactions.

The electrospinning process allows a one-pot synthesis to have the support and metallic particles [5,17,32,33,34,64,65] and does not impact the morphology and the size of particles created during the process [32]. Different metals were tested with PAN in DMF (see Table 2). Figure 2 shows the scanning electron microscopy (SEM) and transmission electron microscopy (TEM) images and different metals on the fibers. This method enabled one to engineer a three-dimensional support with different metals, leading to a high surface area support that can act as a scaffold for enzymes immobilization [41] or catalytic reactions [65]. For Figure 2, electrospun PAN nanofibers were prepared, and the nanoparticles were then loaded onto the carbonized electrospun mat to yield supported high-entropy-alloy nanoparticles (HEA-NPs) [65].

Two analyses are commonly performed to get the information about the fiber quality. By SEM, it is possible to get the information on the size of the particles and the fibers. The coupling to energy-dispersive X-ray spectroscopy (EDX) enables one to get the qualitative data on the nature of chemical species. To get quantitative data (bulk analysis), complementary elementary analysis by inductively coupled plasma (ICP-OES or ICP-MS) for heavier elements and CHNS-O (the determination of C, H, N, S, and O contents) should be done. For thinner samples, transmission electron microscopy (TEM) can be used to reach a nanometer scale information and provide more detail about the sample. SEM is more used than TEM because the samples made by electrospinning could be too thick for TEM analysis, because SEM is more representative of the sample (not a nm information), and because it is less expensive and less time consuming. Some direct analysis by TEM can be carried out [31], but most of the time, the materials are ground into a powder similar to the preparation for the electrocatalytic tests [66,67,68,69,70].

**Table 2 materials-15-04336-t002:** Different types of materials prepared as fibers with PAN via electrospinning.

Material	Solvent	Reference
Ni/N,S-doped carbon	DMF	[48]
CoSe/N-doped carbon	DMF	[71]
Fe_3_O_4_/N-doped carbon	DMF	[49]
WS_2_/N-doped carbon	DMF	[50]
MnCo_2_O_4_@N-doped carbon	DMF	[51]
Co_3_O_4_-CNFs (CNF: carbon nanofibers)	DMF	[72]
Au@CFs (CFs: carbon fibers)	DMF	[34]
Casein/PAN	DMSO	[42]
Li/CNFs	DMF	[35]
NCNFs (nitrogen-doped electrospun carbon nanofibers)	DMF	[45]
PAN NFM (polyacrylonitrile electrospun nanofibrous membrane)	DMF	[41]

To explain the formation of AuNPs during the electrospinning process without extra energy, it was argued that DMF can act as a reducing agent under suitable conditions to convert Au(+III) into Au(0) [30,34]. However, the ability of DMF to reduce metallic cation into the zero oxidation state depends on the nature of the involved metal (AuCl_4_^−^ ions into Au^0^ vs. Ag^+^ into Ag) [30]. Given that DMSO medium is not known to act as a reducing agent and that the AuNPs are formed in the presence of DMF or DMSO, it is argued that PAN polymer can also act as a reducing agent to reduce the Au(+III) salt under mild conditions (70 °C) [34]. However, there is a limitation to this method. The quantity of metal salt added in solution should not exceed a certain percentage depending on the targeted metal because it becomes more difficult to create fiber during electrospinning (the formulated solution is not so homogeneous, see the previous section).

### 2.4. Process of Electrospinning for PAN

A thorough literature analysis shows a certain difference in the used experimental parameters to spin PAN fibers, but in the majority, they are in the same range. One common thing is that the grounded counter electrode is connected to an aluminum foil as a collector for the experiment. For the electrospinning process, the applied voltage ranges from 11 kV [47,52,64] to 80 kV [42]; it should be noted that this parameter depends on the needle-to-collector distance and the composition of the electrospinning solution (see Section 2.1). The most commonly used voltage is 20 kV [34,35,39,40,41,46,53,56] for an average needle-to-collector distance of 15 cm (that is, 100 kV/m as an estimated electric field), whereby the majority of the studies are done for 10–11 wt.% of PAN in DMF [34,39,40,47,48,49,50,51,52]. Salles et al. reported a voltage of 4.5 kV when the distance between the needle tip and the metallic target was 10 cm, that is, an estimated electric field of 45 kV/m [73]. By assuming that it is almost impossible to perform electrospinning below 10 kV for 10–15 cm as the needle-to-collector distance, it is therefore likely that the measurement of this voltage value was underestimated. Another parameter of key importance is the distance between the needle and the collector, which is kept constant through the experiment. The lowest values are 6–10 cm (voltage of 4.5 kV [73] or 12 kV [47]), the highest values are 20–25 cm (voltage of 80 kV [42] or 20–25 kV [56]), and the most used value is 15 cm [34,35,39,40,41,46,53] (voltage of 20 kV). The presence of additives can also lead to a change in the conditions. For example, according to Both Engel et al. [34], 10 mL DMF with 1 g PAN is performed under 20 kV for a tip-to-collector distance of 14 cm (flow rate of 2.4 mL/h at 2000 rpm) while in the presence of 0.5 g HAuCl_4_, and the voltage is increased to 22 kV for a tip-to-collector distance of 8 cm (the drum speed reduced to 1000 rpm for a flow rate of 2 mL/h) because of the change in the solution properties. For the flow rate, the widely used value ranges from 0.075–0.1 mL/h [47,64,73] to 2.4 mL/h [34,39], with an average of 1.5 ± 0.5 mL/h [35,41,46,53,56]. For the needle, the commonly used diameter is 800 µm, and diameters of 1.2 mm [35] and 0.5 mm [64] were also reported. For the drum collector, the rotational speed is 1000 rpm [34] or 2000 rpm [34,39]; unfortunately, many works [41,53] do not specify the value, which compromises the reproducibility of the experiments.

The previous points highlight different experimental conditions for the synthesis of PAN-based electrospun mats. The difference might be directed by the intended application of the fabricated materials. The parameters for one system are not necessarily the same for others (see above). For a given synthesis, the weight percent of the PAN polymer and the addition of metallic salts or organic molecules could impact the optimized parameters shown before. To make sure that high-quality fibers will be collected, a further optimization might be necessary. For example, considering “PAN” and “PAN with gold”, the parameters are different in order to create the mat of fibers [34]: the voltage, needle-to-collector distance, the flow rate, and the collector rotational speed are different. It should be kept in mind that not only operational parameters have an impact on the fiber synthesis but also environmental parameters such as temperature, relative humidity, etc., and, to our knowledge, only one study [42] among all those cited above has provided this key information. For example, electrospinning between summer and winter periods is subject to a temperature difference of up to 30 °C. As a result, the quality of the collected material would not be the same if the parameters are left unchanged, e.g., the humidity level will impact the viscosity. The issue of research reproducibility [74] is inherently linked to the non-disclosure of all experimental conditions, which is very problematic and may arise from the complexity of the studied materials or from a lack of rigor [75,76,77,78]. In an editorial entitled “The Experimental Section: The Key to Longevity of Your Research”, Buriak and Korgel summarized that “one of the greatest compliments anyone can give your published work is to reproduce it and build upon it” [76]. The two identified reasons of irreproducibility in scientific reports are the sheer carelessness and the misguided attempt at obtaining or maintaining a competitive advantage by intentionally withholding critical experimental details; these lead to short-term gains but very profound long-term losses as future papers are viewed with a skeptical eye or simply ignored [76].

Once the electrospinning is properly designed and done, at the end of the process, a mat composed of micro/nanofibers is obtained, as shown in SEM images and the photos of Figure 3 in which the fibers’ diameter ranges from 1 to 5 µm [34]. The yellowish color is obvious for the tissues collected after electrospinning of a solution containing HAuCl_4_. This mat could then undergo thermal or chemical treatment depending on the intended use, which is the purpose of the next section.

### 2.5. Thermal Treatment

Thermal treatment is needed to create CNFs with enhanced electrical conductivity, which is important for electrochemical applications. The step is also needed to consolidate the electrospun mat. For that, two steps are needed. The first step is the stabilization process to make a reticulation of the polymer. After that, the second stage is the carbonization process to make the carbon fibers with a graphitic structure. For the PAN-based materials, extensive studies have been done and the parameters are the same as for the first step. This step is routinely done under air atmosphere at a temperature between 220 °C and 300 °C for a dwell time of 1–3 h [17,34,35,37,39,52,53] in order to trigger cyclization, the dehydrogenation and oxidation process [35,38,52,56,79], as shown in Figure 4**,** which is the evolution of the structure of the PAN molecule during the thermal treatment.

By increasing the temperature under air atmosphere, the color of the PAN fibers goes from white to black, as shown in Figure 5, and is assigned to the formation of a ladder ring structure [35]. The weight loss also increases from 5 wt.% at 235 °C to 15 wt.% at 300 °C [17], as shown in the thermogravimetric analysis (TGA) of Figure 6a. At the same time, the shrinking of the PAN mat diminishes, depending on the temperature and dwelling time. TGA is a thermal analysis technique that consists of measuring the variation in mass of a sample as a function of time, for a given temperature or temperature profile. It is used to determine the degradation temperatures, the moisture absorbed by the material, and the amount of organic and inorganic compounds in a material. Here, it is used to determine the best temperature for the stabilization process to get the full cyclization of the PAN molecule and the weight loss after thermal treatments. It could be combined with differential scanning calorimetry (DSC). This analysis measures the difference in heat exchange between the sample and a reference and allows one to determine the phase transitions, for instance, the glass transition temperature and the melting or crystallization temperature. For PAN, it is possible to determine the best stabilization condition that corresponds to the main peak in the DSC curve; the position ranges from 250 to 290 °C. This step corresponds to the shrinking of the PAN.

After this first step, the carbonization process could be done at a temperature of 600–1400 °C under either argon or nitrogen gas [52,80,81,82]. This step should be done in an inert atmosphere to: (i) avoid the total loss of weight by the complete mineralization and (ii) create an electrically conductive carbon material for use in electrocatalytic tasks. Figure 6a clearly shows that if the second step is done in an O_2_ atmosphere, only CO_2_ will be created, and structures such as Figure 6b,c will not be obtained. This step permits one to increase the C/N ratio and, at the same time, to create an electrically conductive graphitic-like structure. The latter depends on the temperature and the dwell time. As shown in Table 3 by X-ray photoelectron spectroscopy (XPS) analysis, when the temperature increases, the C/N ratio increases (4.8 and 7.6 at 573 K (300 °C) and 1073 K (800 °C, respectively) [80], which impacts the electrical properties of the material [52,80,81]. The type of nitrogen species also changes: the main species are pyridinic-N (N-6), pyrrolic-N (N-5), oxidized-N (N-X), and quaternary-N (N-Q: the exact nature is subject to keen debate) [80].

For PAN electrospun mats with gold, Both Engel et al. [34] have substantially increased the duration of the carbonization step from 1 to 10 h at 1000 °C to study the impact on the formation of nanostructured gold particles inside and outside the graphitized carbon microfibers. SEM images of the sample annealed for 10 h are presented in Figure 7. The main observation is the formation of two particle-sized populations, which was approximately the same result upon annealing for 1 h. The thermal treatment seems to not affect the particles’ size or morphology; only the polymer changes into CNFs, and likely the nitrogen content too [35]. The most widely used temperature for the carbonization of PAN-based materials ranges from 550 to 1200 °C for 1 h [34,35,39,42,53]. Zhang et al. [35] showed that the crystallites’ size increases while the nitrogen content decreases when the carbonization temperature goes from 550 to 950 °C under nitrogen atmosphere; 950 °C (1 h) leads to the optimal electrochemical performance.

### 2.6. Other Characterizations of the Mat Fibers

Other characterizations are possible for the PAN mats before and after any thermal treatment. The Fourier Transform Infrared Spectroscopy (FTIRS) gives information about the structure of the molecule, that is, the type of organic functions (alcohol, amine, etc.). Figure 8a,b shows the FTIR spectra for different types of PAN-based materials. For PAN, the bands at 2940, 2250, and 1450 cm^−1^ correspond to the stretching of C–H, C≡N and CH_2_, respectively [35,40,52,56,79]. After stabilization, their intensities decrease and an additional peak appears at 810 cm^−1^, which is assigned to the vibration of =C–H. The new peak comes from the aromatic ring that is created during the cyclization reaction (see Figure 4) [35,52]. Indeed, after carbonization under N_2_ gas, the C≡N and CH_2_ bands must disappear because of crosslinks between the PAN molecules. Therefore, any presence of a peak at 2250 cm^−1^ suggests that the carbonization step is not yet completed [56]. It was reported that at 500 °C, C=N and C–N bands at 1374 and 1284 cm^−1^, respectively, are strengthened due to the conversion of C≡N, while at 950 °C, the C=N vibration band disappears (other reaction products can be detected depending on the applied temperature) [35,52,56,79].

X-ray diffraction (XRD) can be used for the crystallinity characterization of the polymer and the metallic particles. Figure 8c shows XRD patterns for different electrospun PAN-based materials. We note the presence of a strong diffraction peak centered at 17° and a weak diffraction peak centered at 29°, which correspond to the X-ray diffraction of (100) and (110) of PAN nanofibers, respectively [52,64,79]. By FTIRS, it was observed that the aromatic growth is the major process during the carbonization at a high temperature, and it triggers the formation of a graphite-like structure by eliminating the nitrogen-containing groups. The decrease in nitrogen content could be confirmed by XPS [35]. When the carbonization temperature increases, the intensity of the peaks corresponding to the graphitic carbon increase while that at 17° disappears [35]. In Figure 8c, at 1000 °C, the carbonized PAN material shows a single diffraction peak centered at 25.6° for the (002) crystallographic plane of graphite [35,52]. For the gold species within the electrospun mats (with and without carbonization), Figure 9 shows the XRD patterns. Figure 9a confirms the role of the solvent and PAN on the formation of gold particles. The peaks situated at 38.2°, 44.5°, 65.6°, and 78.6° correspond to, respectively, (111), (200), (220), and (311) planes of face-centered cubic structure of gold [34,47,64,83]. Leveraging on the evolution of the intensity and the width of the (200) and (220) peaks, it was argued that metallic gold particles are created during the electrospinning process and the thermal treatment changes the crystallinity [34].

## 3. HMF Electrooxidation for Paired Electrosynthesis of Valuable Chemicals

In the previous Section 2, we focused on the electrospinning method to obtain materials that can be used as electrocatalysts. The current Section 3 focuses on a summary review of the electrocatalysis of HMF. Ideally, we would like to review the performance of the materials from Section 2; however, as there are no data available, to our knowledge, on the use of electrocatalysts derived from electrospinning for the electroconversion of HMF, we will review other systems to provide some inspirational ideas.

To reduce the H_2_ price by an electrochemical process, few solutions are reported. One of them is to reduce the overpotential of the oxygen evolution reaction (OER) by the electrooxidation of organic molecules to decrease the quantity of energy used to form H_2_ at the cathode of an electrolyzer [6,12,84,85,86,87,88]. It has been showed that by oxidizing at the anode a biomass derivate, it is possible to reduce the required voltage compared to the electrooxidation of water molecules [89]. The techno-economic analysis of renewable energy-powered biomass electrolysis where organics are oxidized at the anode in lieu of water has shown the promise of H_2_ production at the cathode with the possible co-generation of valuable organics at the anode [11,85,89,90]. For example, Paul Kenis’ group reported lower electricity consumption by up to 53% compared to conventional water electrolysis [91]. Figure 10a shows that glucose and glycerol have a much lower oxidation potential than the oxidation of water, so less energy will be required to produce H_2_ by such an electrolysis [6]. We note that the cell voltage of an electrolyzer is *U*_cell_ = *E*_(anode)_ − *E*_(cathode)_. Consequently, the potential of the anode where the electrooxidation is happening must be as small as possible for a given cathode material. As shown in Figure 10b, HMF electrooxidation is less energy-demanding than OER [12], and its electrocatalysis can be regulated to produce value-added chemicals to replace petroleum-based polymers [10,12,13].

The linear sweep voltammetry (LSV) of Figure 10b shows that the presence of 50 mM HMF in 1 M KOH is accompanied by a drastic shift in the potential needed to drive more current density (and thus more conversion of the reactants into products, the second law of Faraday). To reach 50 mA cm^−2^, a potential of ca. 1.42 V vs. RHE (reversible hydrogen electrode) is needed in the presence of HMF, which is 150 mV smaller than that of OER (1.57 V vs. RHE). The goal of using organic molecules at the anodic compartment of a H_2_ production electrolyzer is not to perform the complete electrooxidation into CO_2_ as it seemingly violates the original intention of reduction the carbon footprint [6,92]. Of course, the CO_2_ from the full degradation of organic molecules could be sent back to the cathodic compartment; this is only valid for CO_2_ electrolyzers [91,93,94], it is not a universal strategy. Here, Figure 11 highlights the dreamed scenarios of the electrocatalytic oxidation of HMF, whereby selective electrooxidation could be a solution to produce useful organic molecules. Hence, in addition to the significant decrease of the total energy input (see Figure 10a), this collateral gain at the anodic compartment contributes to the decrease of the cost of H_2_ production (Figure 12a). The two possible pathways without any carbon–carbon bonds cleavage are the aldehyde route (Pathway A) and the alcohol route (Pathway B), which can change depending on the used electrode material (Figure 12). We note that HMF can be conventionally oxidized by using several methods: (i) in hard conditions with organic solvents at high pressure of O_2_ and temperature [7,95,96], (ii) by an enzymatic method [97], and (iii) by an electrocatalytic method using synthetized metal-based electrodes [2,13,16,34,98,99,100,101]. The promise of the breakthrough electroconversion of HMF by electrocatalysis relies on the use of mild conditions (temperature, pressure), only water as an oxygen source, solid electrode materials, and electrons as the triggers (from any renewable electricity source, solar, wind, etc.). Compared to catalysis by oxidants such as O_2_ at elevated temperature and pressure, this set of specifiers contributes to making electrocatalysis a more sustainable process and renders electrochemistry environmentally useful as a “green tool” [102,103,104].

For Pd-Au materials, the required cell voltage (Figure 12b), the reaction paths (Figure 12c,f), and the yield (Figure 12d,e) depend on the exposed active sites at the electrocatalyst surface. Under chronoamperometry condition (0.82 V vs. RHE in 1 M KOH + 5 mM HMF), the conversion of HMF into FDCA is more important at bimetallic Au-Pd in comparison to the monometallic Pd and Au materials [16]. The selectivity improvement can be explained by the synergistic actions between Au and Pd, driven by the coupling of electronic and geometric effects. It can also be observed that Au outperforms Pd for the oxidation of HMF into HFCA (Figure 12e). It is well-known that Au is more active for the oxidation of the aldehyde group, whereas Pt and Pd catalysts are more active for the oxidation of the alcohol groups [98]. When they are combined, the electrooxidation could start at low potentials [13,16,98]. For the cost reduction, it is possible to use non-noble metals. For example, HMF was successfully converted into DFF with 78% isolated yield and 100% selectivity by an environmentally friendly organic electrosynthesis using graphite as an anode and stainless steel as a cathode [2]. You et al. [87] reported 3D hierarchically porous nickel-based electrocatalyst obtained by the electrodeposition to reduce the cell voltage by 220 mV at 50 mA cm^−2^ when compared to conventional water splitting.

The above examples show the variety of opportunities to electrooxidize HMF into different components. Other options exist, starting from fructose because of the possible stability issues of HMF in aqueous alkaline solutions where the aldehyde function could be converted into carboxylic acid by a nucleophilic attack by OH^−^. Control trials are currently missing to fully know the impact of such a hypothesis on the whole mechanism. Indeed, the electrooxidation of any organic molecule is a set of proton-coupled electron (PCET) steps so that the pH has a key impact on the reactivity that is expected to be maximal at pH close to the compound’s p*K*a [105,106,107,108,109,110]. Given the HMF’s p*K*a of 12.8, working in an alkaline electrolyte to maximize electrocatalytic activity exposes HMF to a nucleophilic attack by OH^−^, which will be amplified as the reaction time increases. Therefore, to provide high yield and selectivity, optimization has to be made. For example, the pH impacts the yield/selectivity, the metal has its own selectivity properties, and the temperature or duration of the electrolysis could impact the stability/yield [2,16,96,97]. Furthermore, the technical report of IUPAC indicates that “the simultaneous transfer of more than one electron is highly improbable” [111]. Thus, the electrocatalysts for HMF electroconversion must withstand a cascade of PCET steps, which will induce many reaction intermediates and energy barriers of larger overpotentials [106]. Each of the discussed parameters should be properly optimized, and theoretical calculations (DFT, MD, etc.) can help to better guide the operation of electrocatalytic materials [6,112].

## 4. Conclusions and Perspective

This review discussed the advances of electrospinning by giving an overview of the experimental conditions for reproducible methodologies. The polymer solution formulation, the spinning, the thermal treatment, and the different analyses have been critically reviewed to understand the chemical and structural changes in the polyacrylonitrile (PAN) molecule and how it can be exploited to engineer high-performance electrode materials for electrocatalysis. During the formulation of the polymer solution for the electrospinning process, it is possible to introduce metallic or organic species to change the final materials’ structure and target different electrocatalytic properties. However, the addition of such a chemical species modifies the physico-chemical properties of the electrospinning solution, as well as the operation parameters. The latter are not rationalized as the methodologies change from one report to another one, which calls for further efforts to structure the field and understand the impact on the electrospinning and the calcination steps.

A focused analysis was conducted for the electroconversion of 5-hydroxymethylfurfural (HMF) into added-value chemicals during CO_2_-free and low energy consumption H_2_ production in electrolyzers. For HMF electrooxidation, further studies should be conducted to obtain free-standing electrocatalysts from electrospinning and target a strong anchoring of metals to support. Thermodynamically, it is true that the electrooxidation of HMF instead of water (OER, oxygen evolution reaction) in the anodic compartment of an electrolyzer for H_2_ production is accompanied by a significant saving of the consumed electrical energy. Nonetheless, this is purely conceptual at the moment because the recorded current densities of a few tens of mA/cm^2^ are too far from the industrial needs of at least 0.5 A/cm^2^ [77,113]: the current density-dependent H_2_ rate at 25 °C is ca. *D* (L/h per cm^2^) = 0.45 × *j* (A/cm^2^). In addition, the field of organic electrosynthesis coupled to the production of decarbonized H_2_ should go beyond the hundreds of reports being published each year by simple three-electrode setups. In fact, there is no guarantee that the implementation in real two-electrode and zero-gap electrolyzers will meet the promise. Theoreticians and experimenters are invited to join their forces in order to significantly increase the current densities while preserving the selectivity that is often obtained at the expense of the current density. In terms of the bigger picture, any developed electrocatalyst for the selective electro-conversion of HMF can be a widespread solution in low-energy input power-to-X technologies where hydrogen evolution (HER), CO_2_ reduction (CO2RR), and N_2_ reduction (N2RR) reactions are considered to enable the electrosynthesis of high value-added fuels and/or chemicals with a significantly reduced environmental footprint. For those interested in electrolysis, we end this review by extending an invitation to a recent Perspective by Siegmund et al. [77] on “Crossing the Valley of Death: From Fundamental to Applied Research in Electrolysis”.

## Figures and Tables

**Figure 1 materials-15-04336-f001:**
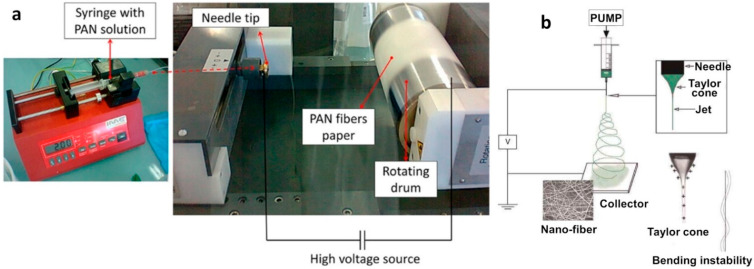
(**a**) Pictures of an electrospinning setup: reprinted and adapted from ref. [54]; Copyright 2015, American Chemical Society, and courtesy provided by Dr. Adriana Both Engel. (**b**) Illustration of the working principle of electrospinning: reprinted and adapted with permission from ref. [55]; Copyright 2015, Elsevier Inc.

**Figure 2 materials-15-04336-f002:**
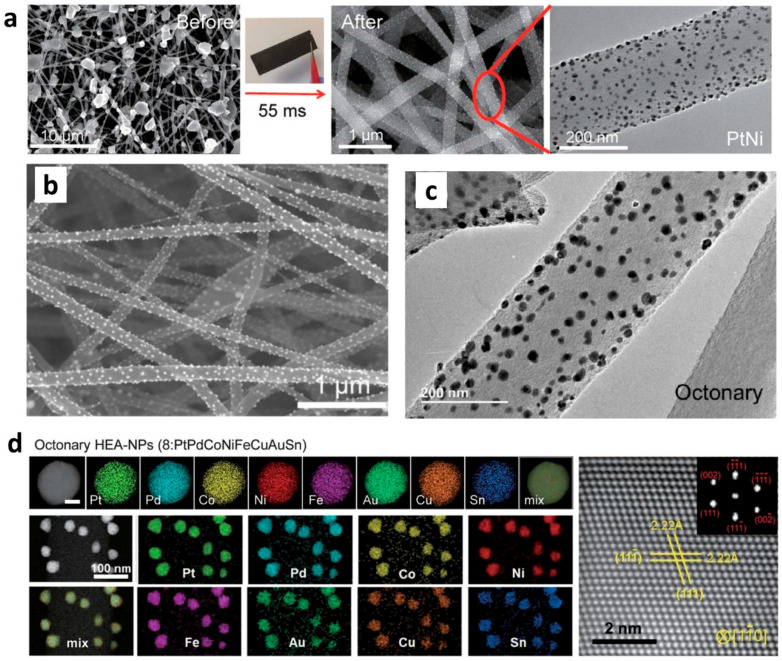
Nanoparticles loading onto carbon nanofibers derived from electrospun PAN nanofibers. (**a**) Illustration of carbothermal shock (CTS) method after electrospinning to synthesize high-entropy-alloy nanoparticles (HEA-NPs): SEM images of microsized precursor salt particles before and after CTS. (**b**–**d**) SEM, TEM, and HAADF-STEM mapping images of octonary alloy nanoparticles. Reprinted and adapted with permission from ref. [65]; Copyright 2018, AAAS.

**Figure 3 materials-15-04336-f003:**
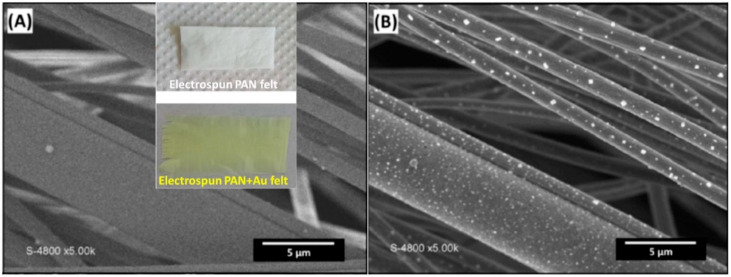
SEM images of Au@PAN fibers (insets are the tissues collected after electrospinning, yellowish for the presence of gold (III) salt) obtained with: (**A**) regular secondary electrons detector and (**B**) back-scattered electrons detector [34]. Reprinted and adapted with permission from ref. [34]; Copyright 2016, Wiley-VCH Verlag GmbH and Co. KGaA, Weinheim.

**Figure 4 materials-15-04336-f004:**
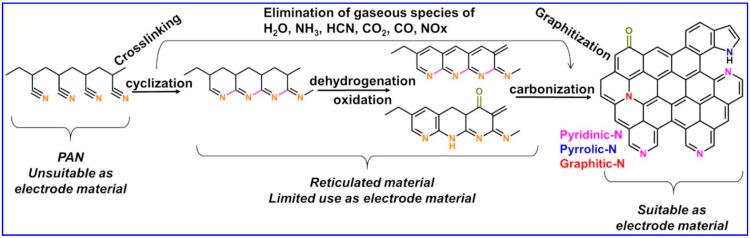
Sketch of the thermal treatment starting from polyacrylonitrile-based material to the fabrication of an electrode material.

**Figure 5 materials-15-04336-f005:**
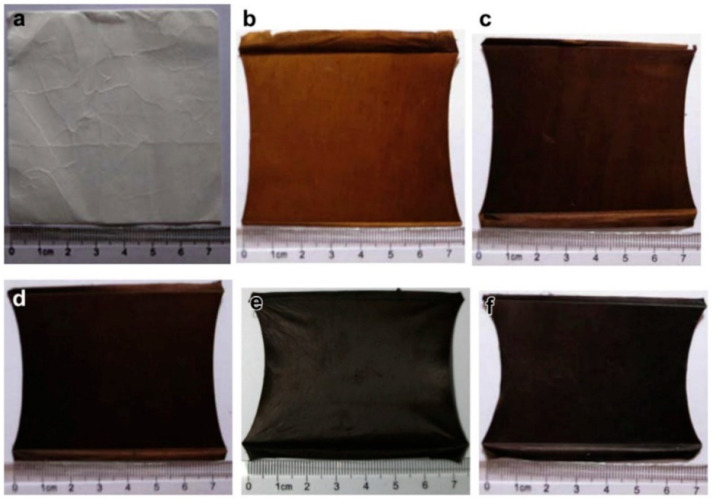
Pictures showing the color change of electrospun PAN mats during stabilization: (**a**) pristine and (**b**–**f**) stabilized (heating rate of 2 °C min^−1^ in a constant air flow, and then maintained at the target temperature): (**b**) 250 °C for 1 h, (**c**) 265 °C for 1 h, (**d**) 280 °C for 1 h; (**e**) 280 °C for 2 h, and (**f**) 280 °C for 3 h. Reprinted and adapted with permission from ref. [52]; Copyrights 2012, Elsevier Ltd.

**Figure 6 materials-15-04336-f006:**
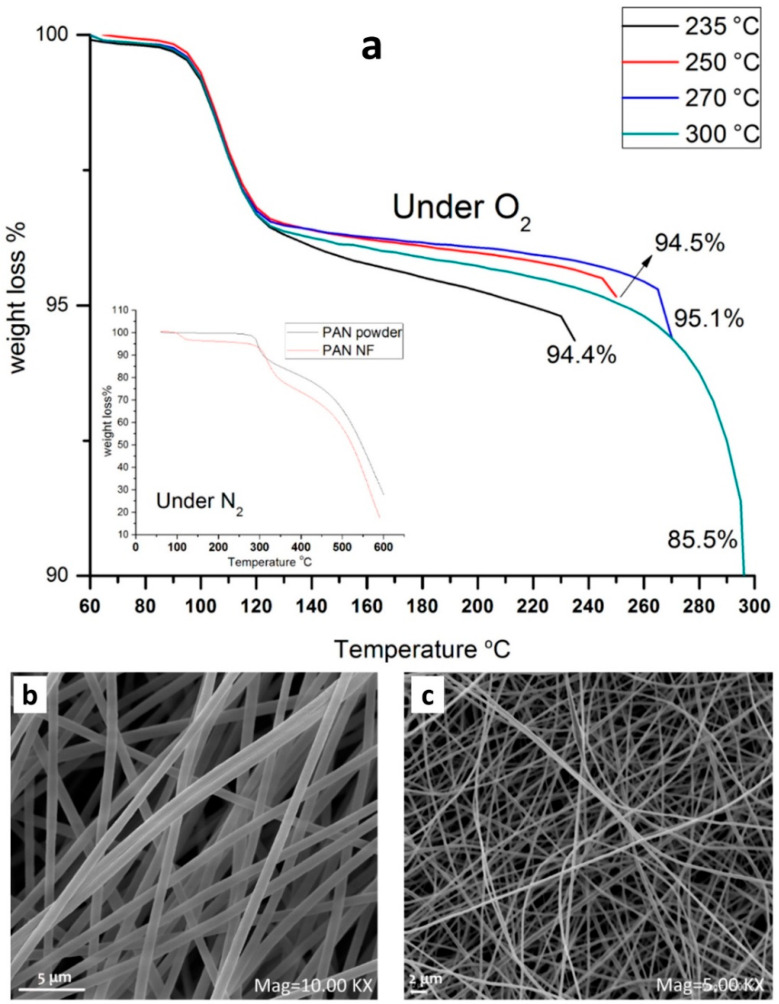
(**a**) The TGA curves of aligned PAN nanofibers (5 °C min^−1^, air) until the targeted oxidation temperature (235, 250, 270, and 300 °C) for a dwell of 6 h (the inset represents the TGA under inert nitrogen atmosphere). SEM images of aligned PAN nanofibers (stabilization at 250 °C for 3 h in air atmosphere and carbonization at 900 °C for 1 h under nitrogen atmosphere) produced with: (**b**) rotating collector and (**c**) with fixed collector. Reprinted and adapted with permission from ref. [53]; Copyrights 2017, Gergin et al.; licensee Beilstein-Institut.

**Figure 7 materials-15-04336-f007:**
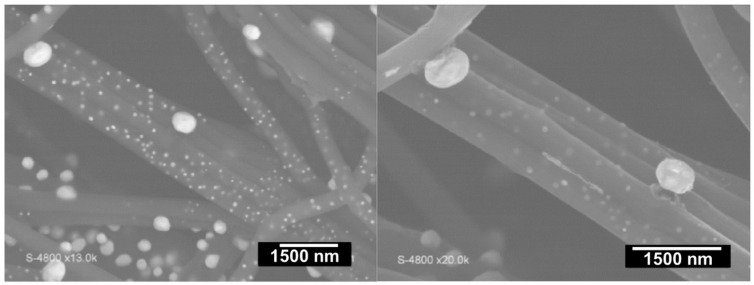
SEM images of Au@CFs at different magnifications after stabilization under air (2 °C min^−1^ until 250 °C for 2 h stay) and calcination under high-purity nitrogen atmosphere (2 °C min^−1^ until 1000 °C for 10 h stay). Reprinted and adapted with permission from ref. [34]; Copyrights 2016, Wiley-VCH Verlag GmbH and Co. KGaA, Weinheim.

**Figure 8 materials-15-04336-f008:**
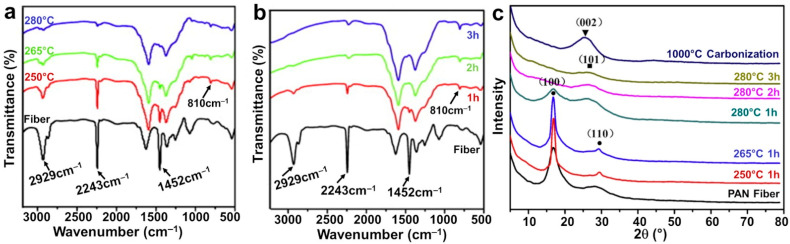
(**a**,**b**) FTIR spectra of stabilized PAN nanofibers (heating rate of 2 °C min^−1^ in a constant air flow): (**a**) temperature effect (holding time: 1 h); (**b**) time effect (holding temperature: 280 °C). (**c**) XRD patterns of the PAN precursor nanofibers, the stabilized (under air), and the carbonized PAN nanofibers (2 °C min^−1^ until 1000 °C for 1 h stay under N_2_ atmosphere). Reprinted and adapted with permission from ref. [52]; Copyrights 2012, Elsevier Ltd.

**Figure 9 materials-15-04336-f009:**
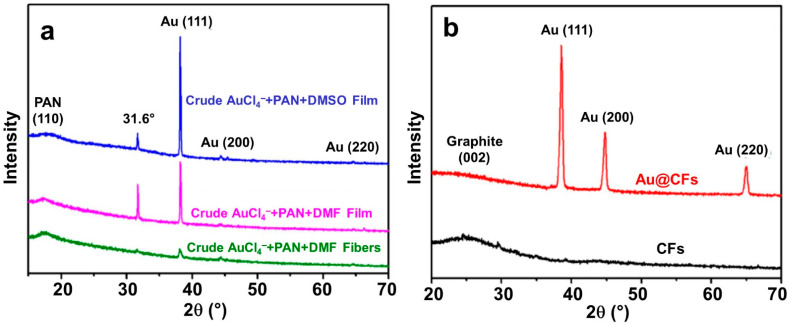
(**a**) XRD of crude fibers obtained from the solution containing AuCl_4_^-^+PAN+DMF (green curve), crude film from the same solution (pink curve), and crude film from the solution prepared in DMSO AuCl_4_^-^ + PAN + DMSO (blue curve). (**b**) XRD of carbonized fibers with no Au (black curve) and Au@CFs (red curve), after stabilization under air (2 °C min^−1^ until 250 °C for 2 h stay) and calcination under high-purity nitrogen atmosphere (2 °C min^−1^ until 1000 °C for 1 h stay). Reprinted and adapted with permission from ref. [34]; Copyrights 2016, Wiley-VCH Verlag GmbH and Co. KGaA, Weinheim.

**Figure 10 materials-15-04336-f010:**
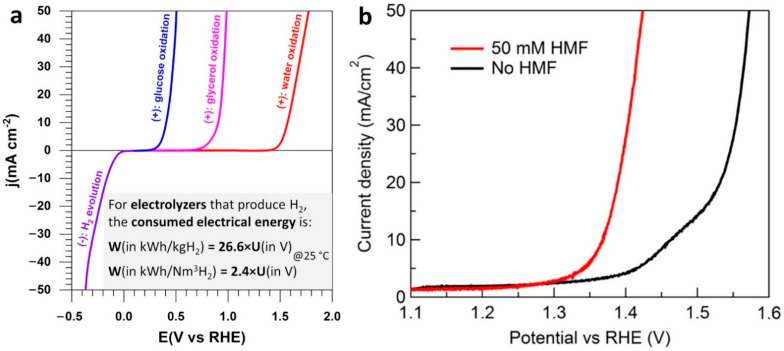
(**a**) Typical half-cell electrolysis polarization curves illustrating the working principle of solid alkaline membrane electrolysis cell [6]. (**b**) LSVs of Co−P/CF (1 M KOH, 2 mV s^−1^) in the absence (black) and presence of 50 mM HMF (red). (**a**) Reprinted and adapted with permission from ref. [6], Copyright 2020, The Royal Society of Chemistry 2020. (**b**) Reprinted and adapted with permission from ref. [12], Copyright 2016, American Chemical Society.

**Figure 11 materials-15-04336-f011:**
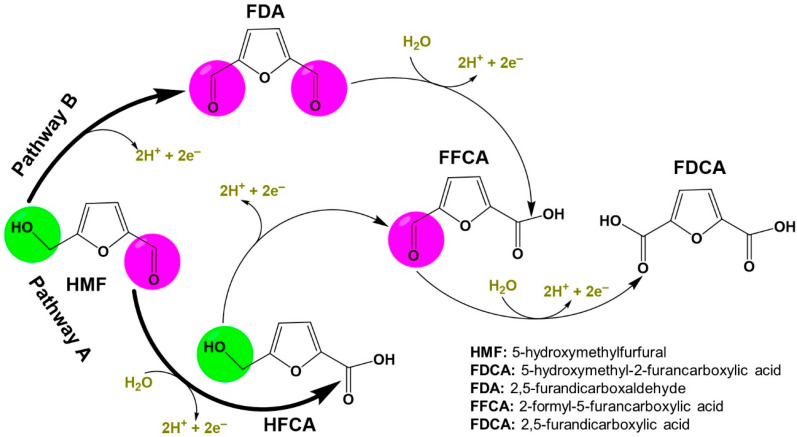
Scenarios of the electrocatalytic oxidation of HMF and the resulting products without C-C bond cleavage (written in acidic media).

**Figure 12 materials-15-04336-f012:**
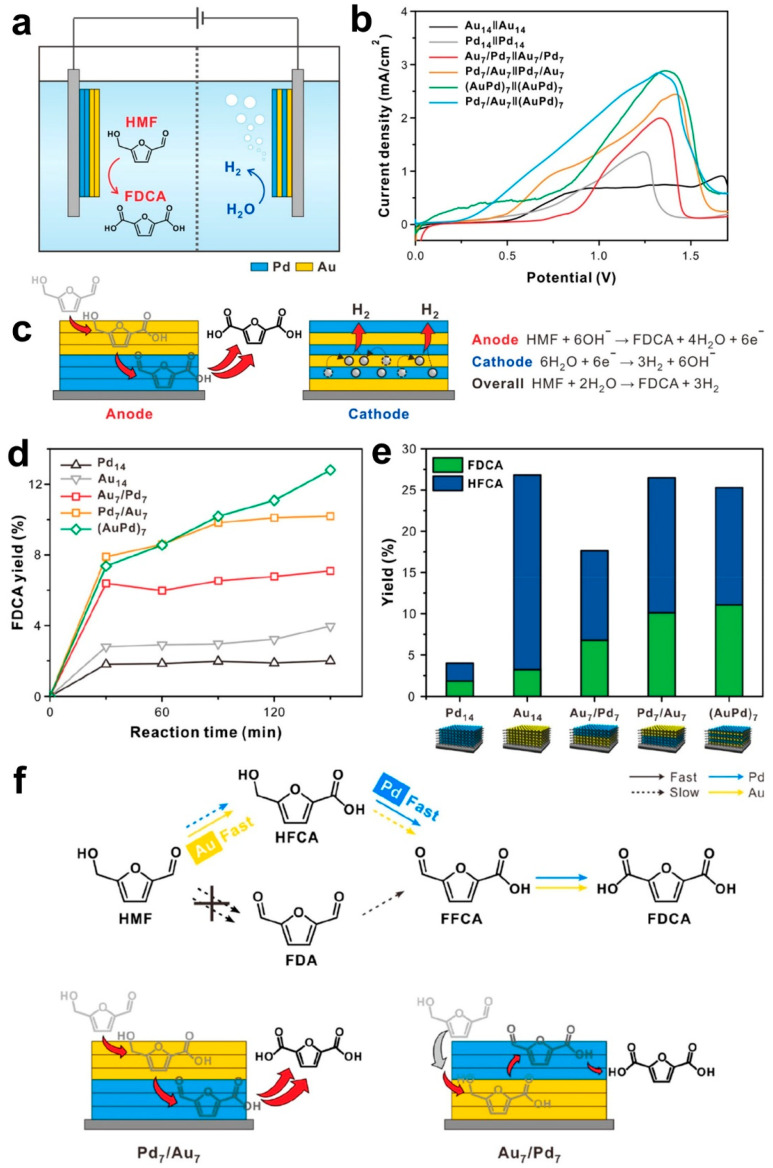
(**a**) Sketch of the two-electrode electrolysis cell made of (+)Pd_7_/Au_7_∥(AuPd)_7_(−) for the simultaneous generation of FDCA from the anode (+) and H_2_ from the cathode (−). (**b**) LSV (2 mV s^−1^) in 1 M KOH + 10 mM HMF. (**c**) Pd_7_/Au_7_ multilayer anode and the (AuPd)_7_ multilayer cathode with the corresponding electron transfer processes. Bulk electrolysis at 0.82 V vs. RHE in 1 M KOH + 5 mM HMF for 2 h at different anode materials. (**d**) Yield of FDCA and (**e**) comparison between HFCA and FDCA. (**f**) Pathways of HMF electrooxidation. Reprinted and adapted with permission from ref. [16], Copyright 2020, American Chemical Society.

**Table 1 materials-15-04336-t001:** Examples of electrospun (bio)polymer materials and the used solvents (in alphabetical order). Reprinted and adapted with permission from ref. [55]; Copyright 2015, Elsevier Inc.

Materials	Solvents
Acrylonitrile-butadiene-styrene (ABS)	N,N-Dimethyl formamide (DMF) or tetrahydrofuran (THF)
Cellulose	Ethylene diamine
Cellulose acetate	Dimethylacetamide (DMAc) and acetone or acetic acid
Ethyl-cyanoethyl cellulose [(E-CE)C]	THF
Chitosan and chitin	1,1,1,3,3,3-hexafluoro-2-propanol (HFIP)
Dextran	Water, dimethyl sulfoxide (DMSO)/water, DMSO/DMF
Gelatine	2,2,2-Trifluoroethanol
Nylon	Formic acid
Poly(2-acrylamido-2-methyl-1-propane sulfonic acid) (PMAPS)	Ethanol/water
Polyacrylonitrile (PAN)	DMF
Polyalkyl methacrylate (PMMA)	Toluene/DMF
Polycarbonate	THF/DMF
Poly(ethylene oxide) (PEO)	Water, ethanol, DMF
Polyethylene terephthalate (PET)	Trifluoroacetic acid (TFA) and dichloromethane (DCM)
Polylactic based polymers	Chloroform, HFIP, and DCM
Pol(ε-caprolacone) based polymers	Acetone, acetone/THF, chloroform/DMF, DCM/methanol, chloroform/methanol, and THF/acetone
Poly(3-hydroxybutyrate-co-3-hydroxyvalerate) (PHBV)	2,2,2-Trifluoroethanol
Polyphosphazenes	Chloroform
Polystyrene	1,2 Dichloroethane, DMF, ethylacetate, methylethylketone (MEK), and THF
Bisphenol-A Polysulfone	DMAc/acetone
Polyurethane (PU)	THF/DMF
Polyvinyl alcohol (PVA)	Water
Polyvinyl chloride (PVC)	DMF, DMF/THF
Poly(vinylidene fluoride) (PVDF)	DMF/THF
Poly(vinyl pyrrolidone)	Ethanol, DCM, and DMF
Silk	Hexafluoroacetone (HFA), HFIP, and formic acid

**Table 3 materials-15-04336-t003:** XPS data from calcined PAN at different temperatures (heating rate of 1 °C/min, dwell time of 1 h, and Ar atmosphere). Reprinted and adapted with permission from ref. [80]; Copyright 1995, Elsevier Science Ltd.

Precursor	Pyrolysis Temperature	Atomic Percentage			Percentage of Nitrogen Functionalities (from XPS N1s)			
		C	N	O	Pyridinic-N (N-6)	Pyrrolic-N (N-5)	Quaternary-N (N-Q)	Oxidized-N (N-X)
PAN	573 K (300 °C)	79.5	16.2	4.3	100	0	0	0
PAN	773 K (500 °C)	78.4	15.5	6.1	69	19	8	4
PAN	1073 K (800 °C)	85.4	11.2	3.5	40	29	23	9

## Data Availability

Not applicable.

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
