# Peer review of "Review of the Electrospinning Process and the Electro-Conversion of 5-Hydroxymethylfurfural (HMF) into Added-Value Chemicals"

_materials, 2022, doi:10.3390/ma15124336_

Round 1
Reviewer 1 Report
- Rewrite the title based on the content of lines 95-100 of the article.
- Please re-extract the abstract of the article according to the content of the article. In my opinion, the content written in lines 10 to 19 can be deleted.
- In the introduction part, when citing the opinion of Professor Peter William Atkins, it should not be quoted directly, but should be quoted indirectly, that is, to extract his point of view based on this sentence. 4. No need to explain, it seems repetitive. For example "Twenty years ago (2002)" etc.
- Please elaborate on the meaning of this sentence in lines 110-112 in the text. "It is worth noting that DMF can act as a reducing agent under suitable conditions and lead to the creation of metal nanoparticles [30]", I think this sentence is wrong.
- When discussing "Formulation of a Suitable Electrospinning Solution" in Section 2.1, in addition to the type of materials, I think the most important thing is the concentration or ratio of the solution, please add relevant content. The comparison of temperature can only be discussed in the case of specific materials and ratios.
- Please standardize the design of Table 1. For example, add a column of "References" to each row in the table. It cannot have only abbreviations without full names, or only full names without abbreviations. Please standardize them uniformly. In addition, it is recommended to unify or merge Table 1 and Table 2. Tables or pictures that reference others cannot be copied directly, and need to be further sorted and processed. Please refer to the chart format in the published paper in Materials magazine
- Please revise the format of lines 157-162 and 168-176. It is recommended to revise them into paragraphs.
- In the subsection 2.4 Process of Electrospinning for PAN, I think the comparison of voltage or distance is meaningless, because neither the concentration of the solution nor the type of solvent in the reference you cited in this paragraph is given here.
- The conclusion needs to be rewritten.
Author Response
Reply to Reviewers’ comments
We would like to thank the reviewers for the careful reading of our manuscript and for the valuable comments. We agree that the comments were relevant, and the critical suggestions brought by the reviewers have been taken into account. You will find below our responses to reviewers’ comments: some paragraphs of the revised manuscript were modified for clarity, highlighted with the red color. Below, we give our response to the comments, as listed by the Reviewers.
Rebuttals to comments of Reviewer # 1
- Rewrite the title based on the content of lines 95-100 of the article.
Author reply. We thank the Reviewer for this suggestion. We have revised the Title: “Review on the Electrospinning Process and the Electroconversion of 5-hydroxymethylfurfural (HMF) into Added-Value Chemicals”.
- Please re-extract the abstract of the article according to the content of the article. In my opinion, the content written in lines 10 to 19 can be deleted.
Author reply. We thank the Reviewer for this suggestion. We have revised the Abstract, however, we have kept some elements to clarify the general context.
- In the introduction part, when citing the opinion of Professor Peter William Atkins, it should not be quoted directly, but should be quoted indirectly, that is, to extract his point of view based on this sentence. 4. No need to explain, it seems repetitive. For example "Twenty years ago (2002)" etc.
Author reply. We thank the Reviewer for this suggestion. We have revised the section.
- Please elaborate on the meaning of this sentence in lines 110-112 in the text. "It is worth noting that DMF can act as a reducing agent under suitable conditions and lead to the creation of metal nanoparticles [30]", I think this sentence is wrong.
Author reply. We have revised the sentence for clarity. Indeed, Ref. 30 (Pastoriza-Santos, I.; Liz-Marzán, L. M., N,N-Dimethylformamide as a Reaction Medium for Metal Nanoparticle Synthesis. 629 Adv. Funct. Mater. 2009, 19 (5), 679-688) presents conditions where DMF can be used for the reduction of metal salts, mainly Au and Ag, while also acting as a solvent. In our group, we have also observed the formation of nanoparticles of gold using DMF.
- When discussing "Formulation of a Suitable Electrospinning Solution" in Section 2.1, in addition to the type of materials, I think the most important thing is the concentration or ratio of the solution, please add relevant content. The comparison of temperature can only be discussed in the case of specific materials and ratios.
Author reply. We thank the Reviewer for this suggestion. We have revised the section.
- Please standardize the design of Table 1. For example, add a column of "References" to each row in the table. It cannot have only abbreviations without full names, or only full names without abbreviations. Please standardize them uniformly. In addition, it is recommended to unify or merge Table 1 and Table 2. Tables or pictures that reference others cannot be copied directly, and need to be further sorted and processed. Please refer to the chart format in the published paper in Materials magazine
Author reply. We understand the reviewer’s concern; we have revised the relevant section wherever possible. For Table 1, we have included the full name, but we cannot add a “References” column because it is a reprint. Also, we think that it is not necessary to put the full name again when it has already been given in the previous lines.
- Please revise the format of lines 157-162 and 168-176. It is recommended to revise them into paragraphs.
Author reply. We thank the Reviewer for this suggestion. We have revised this section.
- In the subsection 2.4 Process of Electrospinning for PAN, I think the comparison of voltage or distance is meaningless, because neither the concentration of the solution nor the type of solvent in the reference you cited in this paragraph is given here.
Author reply. Yes, we agree with the Reviewer that the voltage should always be linked to the distance between the needle and the collector and all that parameters are linked to the composition of the electrospinning solution. We have revised the sentence.
- The conclusion needs to be rewritten.
Author reply. We thank the Reviewer for this suggestion. We have revised the conclusion section.
Reviewer 2 Report
Coronas et al. have presented an amazing review focusing on the electrospinning by examining a number of know-hows of experimental conditions. Electrospinning is a fiber-spinning technology used to produce three-dimensional and ultrafine fibers with tunable diameters and lengths. The thermal treatment and the different analyses are discussed to understand the changes in the polymer to create usable electrode materials. It seems that the review is of great interest to Materials readers. In fact, the aim of this review is to summarize the latest development in the used of the electrospinning to engineer advanced electrocatalytic materials for electrosynthesis tasks. The manuscript is informative, well organized, and rich of relevant references and innovative insights. Therefore, it can be accepted after minor revision since there are no technical/major comments. The reviewer's comments are below:
1) Some close reviews dealing with the same topic have already been reported in the literature. I recommend to emphasize the main additional contribution in this review as well as the difference between them.
2) The manuscript should be proofread to correct some errors and typos. The English should be improved. (e.g., page 2, line 195, The aim of this review is to summarize the latest development in the used of the electrospinning to engineer advanced electrocatalytic materials). I think that the indicated sentence should reformulated.
3) I recommend to improve the quality of some figures such as Fig. 1b; Fig. 4; Fig. 6a.
Good luck.
Author Response
Reply to Reviewers’ comments
We would like to thank the reviewers for the careful reading of our manuscript and for the valuable comments. We agree that the comments were relevant, and the critical suggestions brought by the reviewers have been taken into account. You will find below our responses to reviewers’ comments: some paragraphs of the revised manuscript were modified for clarity, highlighted with the red color. Below, we give our response to the comments, as listed by the Reviewers.
Rebuttals to comments of Reviewer # 2
Coronas et al. have presented an amazing review focusing on the electrospinning by examining a number of know-hows of experimental conditions. Electrospinning is a fiber-spinning technology used to produce three-dimensional and ultrafine fibers with tuneable diameters and lengths. The thermal treatment and the different analyses are discussed to understand the changes in the polymer to create usable electrode materials. It seems that the review is of great interest to Materials readers. In fact, the aim of this review is to summarize the latest development in the used of the electrospinning to engineer advanced electrocatalytic materials for electrosynthesis tasks. The manuscript is informative, well organized, and rich of relevant references and innovative insights. Therefore, it can be accepted after minor revision since there are no technical/major comments. The reviewer's comments are below:
1) Some close reviews dealing with the same topic have already been reported in the literature. I recommend to emphasize the main additional contribution in this review as well as the difference between them.
Author reply. We agree with the Reviewer that the same topic has been already reported in the literature. Our main contribution is to examine step-by-step a number of know-hows of experimental conditions from electrospinning until obtaining electrically conductive carbonaceous material. So, we firstly examined the roles of different experimental parameters for the electrospinning process, from the formulation of the polymer solution to the thermal treatments to recover a suitable material. Our review also diagnoses the recently designed strategies during the last years for the electrochemical conversion of the biomass derivate, HMF, into added-value chemicals in aqueous media.
2) The manuscript should be proofread to correct some errors and typos. The English should be improved. (e.g., page 2, line 195, The aim of this review is to summarize the latest development in the used of the electrospinning to engineer advanced electrocatalytic materials). I think that the indicated sentence should reformulated.
Author reply. We thank the Reviewer and we sincerely apologize for these typos issues. We have double-checked the entire manuscript.
3) I recommend to improve the quality of some figures such as Fig. 1b; Fig. 4; Fig. 6a.
Author reply. The resolution of Figures 1b, 4 and 6a has been improved.
Reviewer 3 Report
The manuscript presents a review of electrospinning and electrospun PAN based nanofiber mats and of the use of 5-hydroxymethylfurfural (HMF) for electrochemical applications. The manuscript starts by discussing the synthesis of nanofibers using PAN, the formation of the precursor solutions, the heat treatments and processing parameters, the addition of gold nanoparticles and their application as electrode in electrocatalysis. Then a discussion of the production of hydrogen and other chemicals by using the HMF electrooxidation was done. A detailed presentation of the recent results and trends in these subjects was made, with up to date references. The review presented in the manuscript is original and only needs minor revisions. I have the following comments:
- On page 2 where it is written “and it is quiet impossible to dope the interior of the fibers” should be “and it is quite impossible to dope the interior of the fibers”
- On page 4 it is written “a direct current (DC) voltage”, which is correct, but confusing. I would be better to put simply “a DC voltage”.
- On page 4 the unit rpm is defined as “round per minute”. It should be “revolutions per minute”.
- Acronyms should be defined before they are used. The manuscript should be carefully revised to check the acronyms. For example, CHNS is used on page 6, but it is not defined in the manuscript.
- On page 7 it is written “Salles et al. reported a voltage of 4.5 kV [73]. By assuming that it is almost impossible to perform electrospinning below 10 kV, it is therefore likely that the measurement of this voltage value was underestimated”. However, for electrospinning, it is the electric field (not the voltage by itself) that is relevant. For small distances voltages can be lower and able to attain the needed high electric field.
- On the end of page 8 and beginning of page 9 it is written “Once the electrospinning is properly designed and done, at the end of the process, a mat composed of micro/nanofibers is obtained as shown in SEM images and the photos of Figure 3 in which fibers ranges from 1 to 5 µm”. Is this range regarding the lengths ? The diameters ?
- On page 9 it is written “The weight loss also increases form 5 wt.%” and should be “The weight loss also increases from 5 wt.%”.
- On page 12 it is written “These two peaks represented the X-ray reflections of (100) and (110)”. XRD peaks are due to diffraction, not reflection.
Author Response
Reply to Reviewers’ comments
We would like to thank the reviewers for the careful reading of our manuscript and for the valuable comments. We agree that the comments were relevant, and the critical suggestions brought by the reviewers have been taken into account. You will find below our responses to reviewers’ comments: some paragraphs of the revised manuscript were modified for clarity, highlighted with the red color. Below, we give our response to the comments, as listed by the Reviewers.
Rebuttals to comments of Reviewer # 3
The manuscript presents a review of electrospinning and electrospun PAN based nanofiber mats and of the use of 5-hydroxymethylfurfural (HMF) for electrochemical applications. The manuscript starts by discussing the synthesis of nanofibers using PAN, the formation of the precursor solutions, the heat treatments and processing parameters, the addition of gold nanoparticles and their application as electrode in electrocatalysis. Then a discussion of the production of hydrogen and other chemicals by using the HMF electrooxidation was done. A detailed presentation of the recent results and trends in these subjects was made, with up to date references. The review presented in the manuscript is original and only needs minor revisions. I have the following comments:
- On page 2 where it is written “and it is quiet impossible to dope the interior of the fibers” should be “and it is quite impossible to dope the interior of the fibers”
Author reply. We sincerely apologize for these typos issues. We have double-checked the entire manuscript.
- On page 4 it is written “a direct current (DC) voltage”, which is correct, but confusing. I would be better to put simply “a DC voltage”.
Author reply. We have updated the sentence to avoid any misunderstanding.
- On page 4 the unit rpm is defined as “round per minute”. It should be “revolutions per minute”.
Author reply. We thank the Reviewer and we have corrected the wording.
- Acronyms should be defined before they are used. The manuscript should be carefully revised to check the acronyms. For example, CHNS is used on page 6, but it is not defined in the manuscript.
Author reply. We have update the manuscript to clarify the meaning, the CHNS-O elemental analysis enables the determination of C, H, N, S and O contents.
- On page 7 it is written “Salles et al. reported a voltage of 4.5 kV [73]. By assuming that it is almost impossible to perform electrospinning below 10 kV, it is therefore likely that the measurement of this voltage value was underestimated”. However, for electrospinning, it is the electric field (not the voltage by itself) that is relevant. For small distances voltages can be lower and able to attain the needed high electric field.
Author reply. Yes, we agree with the Reviewer that the voltage should always be linked to the distance between the needle and the collector. We have revised the sentence. Indeed, in this reference, one can read 10 cm in the SI while 6 cm is mentioned in the main text, which add a complexity.
- On the end of page 8 and beginning of page 9 it is written “Once the electrospinning is properly designed and done, at the end of the process, a mat composed of micro/nanofibers is obtained as shown in SEM images and the photos of Figure 3 in which fibers ranges from 1 to 5 µm”. Is this range regarding the lengths ? The diameters ?
Author reply. It is about the diameter of the fibers; the sentence was corrected.
- On page 9 it is written “The weight loss also increases form 5 wt.%” and should be “The weight loss also increases from 5 wt.%”.
Author reply. We sincerely apologize for these typos issues. We have double-checked the entire manuscript.
- On page 12 it is written “These two peaks represented the X-ray reflections of (100) and (110)”. XRD peaks are due to diffraction, not reflection.
Author reply. We have updated the sentence to correct the mistake.
Reviewer 4 Report
The topics raised in the article are very relevant, but very little new data is presented in this review.
The data described in the Section 2, devoted to the production of PAN fibers, their transformation into CF fibers, and characterization, has been repeatedly described in many reviews. Therefore, this part of the review contains almost no novelty except for some phrases about Au particles' influence on the XRD.
The work does not trace a coherent and connected presentation between sections 2 and 3.
The structure of the article should be stated clearly.
Line 169 Here should be added such an important parameter as a polymer-solvent affinity and can be estimated by Hansen method or direct measurements like static light scattering or classical capillary viscometry with obtaining of Huggins constant and intrinsic viscosity values
Line 290 etc. I can only agree with the problem described there. Unfortunately, it is a common situation now, there experiments cannot be represented by research published, moreover, sometimes it is written absolutely wrong data or synthesis/spinning methods.
Lines 326, 332, 379. I think there is no need to explain what TGA, DSC, and FTIRS measure for the high rating journal.
Some minor flaws:
Line 245 The phase “to synthesize PAN fibers” is incorrect, and should be replaced with “to spin PAN fibers”
There are some stilistic errors, for example, hyphen is written instead minus sign, etc.
Author Response
Reply to Reviewers’ comments
We would like to thank the reviewers for the careful reading of our manuscript and for the valuable comments. We agree that the comments were relevant, and the critical suggestions brought by the reviewers have been taken into account. You will find below our responses to reviewers’ comments: some paragraphs of the revised manuscript were modified for clarity, highlighted with the red color. Below, we give our response to the comments, as listed by the Reviewers.
Rebuttals to comments of Reviewer # 4
The topics raised in the article are very relevant, but very little new data is presented in this review.
The data described in the Section 2, devoted to the production of PAN fibers, their transformation into CF fibers, and characterization, has been repeatedly described in many reviews. Therefore, this part of the review contains almost no novelty except for some phrases about Au particles' influence on the XRD. The work does not trace a coherent and connected presentation between sections 2 and 3. The structure of the article should be stated clearly.
Author reply. We agree with the Reviewer that the same topic has been already reported in the literature. Our main contribution is to examine step-by-step a number of know-hows of experimental conditions from electrospinning until obtaining electrically conductive carbonaceous material. So, we firstly examined the roles of different experimental parameters for the electrospinning process, from the formulation of the polymer solution to the thermal treatments to recover a suitable material. Our review also diagnoses the recently designed strategies during the last years for the electrochemical conversion of the biomass derivate, HMF, into added-value chemicals in aqueous media. We have revised the Section 3 to clarify the structure. Indeed, in Section 2, we focused on the electrospinning method to obtain materials that can be used as electrocatalysts. The Section 3 focuses on a summary review of the electrocatalysis of HMF over the last ten years. Ideally, we would like to review the performance of the materials from Section 2, however, as there are no data available, to our knowledge, on the use of electrocatalysts derived from electrospinning for the electroconversion of HMF, we will review other systems with a perspective of providing some inspirational ideas.
Line 169 Here should be added such an important parameter as a polymer-solvent affinity and can be estimated by Hansen method or direct measurements like static light scattering or classical capillary viscometry with obtaining of Huggins constant and intrinsic viscosity values
Author reply. We thank the Reviewer for this suggestion that has been taken into account.
Line 290 etc. I can only agree with the problem described there. Unfortunately, it is a common situation now, there experiments cannot be represented by research published, moreover, sometimes it is written absolutely wrong data or synthesis/spinning methods.
Author reply. We thank the Reviewer for agreeing with us.
Lines 326, 332, 379. I think there is no need to explain what TGA, DSC, and FTIRS measure for the high rating journal.
Author reply. We thank the Reviewer, however, we consider that including this brief description allows those who are not familiar with these techniques to understand more about the interpretation of the data.
Some minor flaws:
Line 245 The phase “to synthesize PAN fibers” is incorrect, and should be replaced with “to spin PAN fibers”
Author reply. We thank the Reviewer for this suggestion. We have revised the sentence.
There are some stylistic errors, for example, hyphen is written instead minus sign, etc.:
Author reply. We thank the Reviewer and we sincerely apologize for these typos issues. We have double-checked the entire manuscript.
Round 2
Reviewer 1 Report
Table 1 and 2 are best in the same format.